# Research Progress of Intelligent Polymer Plugging Materials

**DOI:** 10.3390/molecules28072975

**Published:** 2023-03-27

**Authors:** Yi Pan, Xianglong Cui, Hao Wang, Xu Lou, Shuangchun Yang, Famuwagun Felix Oluwabusuyi

**Affiliations:** 1Department of Petroleum and Natural Gas Engineering College, Liaoning Petrochemical University, No. 1, West Section of Dandong Road, Wanghua District, Fushun 113001, China; 2Department of the Institute of International Education, Liaoning Petrochemical University, Fushun 113001, China

**Keywords:** polymer, gel, reservoir, leakage, collapse, gap, plugging

## Abstract

Intelligent polymers have become the focus of attention worldwide. Intelligent polymer materials through organic synthesis methods are used to make inanimate organic materials become “feeling” and “sentient”. Intelligent polymer materials have been applied in actual engineering production, and they are becoming a new research topic for scientists in various fields and countries, especially in the areas of drilling and plugging. The development of intelligent polymer materials can provide new solutions and technical means for drilling and plugging. Unlike traditional plugging materials, intelligent polymer plugging materials can cope with environmental changes. They have the characteristics of a strong target, good plugging effect, and no damage to the reservoir. However, there are currently no reviews on intelligent polymer plugging materials in the drilling field, so this paper fills that gap by reviewing the research progress of intelligent polymer plugging materials. In addition, this paper describes the mechanism and application status of intelligent polymer shape-memory polymers, intelligent polymer gels, intelligent polymer membranes, and intelligent polymer bionic materials in drilling and plugging. It is also pointed out that some intelligent polymer plugging materials still have problems, such as insufficient toughness and a poor resistance to salt and high temperature. At the same time, some suggestions for future research directions are also presented for reference.

## 1. Introduction

Intelligent polymer materials (IPMs) are also called stimulus-responsive polymers. They can sense changes in the surrounding environment and take relevant response measures. They can self-learn, proliferate, and discriminate stimuli. As early as 1970, Toyoichi Tanaka in Japan discovered the phenomenon of intelligent polymers. When the polyacrylamide gel is cooled, the gel will gradually become cloudy and eventually opaque, and when heated, it turns transparent again [1]. In the 1980s, intelligent polymer materials used to make polymer sensors, separation membranes, and artificial organs emerged. Intelligent polymers entered a phase of rapid development in the 1990s. After the 21st century, intelligent polymer materials are developing in the direction of intelligent polymer fuzzy materials. Mitsubishi Heavy Industries (MHI) successfully developed thermoplastic polyurethane shape-memory polymers (SMPURs) in 1988. It has a wide range of Tg, and its processability is superior to previous SMP materials [2,3]. Therefore, these SMPs have received much attention and have a good application prospect. By 2000, IPM research was already involved in various fields such as biomedicine, aerospace, and petroleum engineering [4,5,6]. Polymer-derived nano-ceramic-metal composites have good wear resistance potential. Sachin Kumar et al. successfully developed PMHS-dispersion Al–Mg–Si alloy nanocomposites by using the FSP technique, which resulted in a 67.2% reduction in wear volume loss and improved wear resistance and delamination resistance of PDC-MMC [7]. Fitriani et al. reviewed the application of polymer films with good flexibility, strength, and temperature resistance properties for packaging materials [8]. Muhammad Rizal Muhammad Asyraf et al. reviewed recent studies on the mechanical properties of sugar palm lignocellulosic fibers with various chemical treatments to assess their potential in structural applications [9].

In petroleum engineering, the drilling process often encounters some complex geological conditions, such as high temperature and high pressure [10]. One of the main problems is well leakage, which often causes hundreds of millions of dollars in economic losses [11]. Therefore, using some plugging materials is the most effective way to solve the well leakage problem, and it can usually achieve a better plugging effect. For example, Santos et al. of Louisiana State University in the United States developed the “diversion material”, and the material “temporarily” expanded to plug the existing fractures when it was pumped into the formation [12]. The fracturing energy eventually focused on the new fracture strands. Then, the bio-degradation and chemical dissolution technology was used to restore the “flow” of those isolated fractures at the end of the plugging process, which effectively protected the formation of secondary contamination. The new curable plugging agent developed by the Southwest Petroleum University has good thixotropy, strong curing properties, and a pressure-bearing capacity of up to 10 MP [13]. It has good compatibility with drilling fluids, can improve the safety of exploration, and has good application prospects. With the development of intelligence in the petroleum industry, intelligent polymer plugging materials have become a hotspot for the drilling engineering [14]. The conventional plugging materials have poor compatibility with oil-based drilling fluids, are unable improve the quality of the filter cake, and have an insufficient formation pressure-bearing capacity. We have carried out exploratory research on the problems of large particle size, insufficient adaptability to leakage channels, and the unsatisfactory plugging effect of shale formations plugged by conventional plugging materials [15].

In recent years, researchers have applied intelligent materials to drilling fluids for leak prevention and plugging, and have developed intelligent shape-memory materials, intelligent gel materials, intelligent molecular membrane materials, intelligent bionic materials, and so on. This work elaborates on the mechanism and application status of intelligent materials, such as intelligent shape-memory polymers, intelligent gels, intelligent membranes, and intelligent biomimetic materials in drilling fluids. According to the mechanism and characteristics of different intelligent materials in drilling fluid, the feasibility and technical approaches of their use in drilling fluid plugging are discussed. It is also pointed out that some intelligent polymer plugging materials still have problems, such as insufficient toughness and a poor resistance to salt and high temperature. At the same time, some suggestions for future research directions are also presented for reference.

## 2. Intelligent Polymer Shape-Memory Polymer

The intelligent shape-memory polymer plugging material can be designed in a curled state or with a curled alloy as the core and a plastic material as the shell. Under normal temperatures, the particle size of the intelligent shape-memory polymer plugging material is small. When it enters the leaky layer with drilling fluid, the temperature inside the layer reaches the phase change temperature of the polymer, and the shape-memory polymer in curled state will be deformed. The increasing radius of curvature will break the shell and the polymers will interpenetrate to form a net. The broken shell will be embedded in the net as plugging particles, thus forming a strong structural accumulation in the leakage channel to achieve the purpose of sealing the leakage layer.

In recent years, the research on intelligent polymer plugging materials have made great progress and play an important role in oilfield production [16]. Intelligent polymer shape-memory polymer (IPSMP) makes up for the defects of conventional plugging materials, such as large particle size, easy-to-block drilling tools, damage reservoir, sealing failure, and wellbore strengthening ability [17,18]. IPSMP is a special polymer material. It is prepared by combining polymer molecular chains [19]. It is a special intelligent material that can remember the morphology of its chains, and it can respond to the influence of external factors (such as heat, electricity, light, magnetism, pH change, etc.) [20]. Usually, this material can recover deformation by up to 10–50% and up to 800%, and this material is gradually becoming the focus of research on intelligent polymer plugging materials [21].

Mansour et al. [22,23,24] conducted a series of studies on IPSMP. The numerical simulations showed a sealing pressure of 35 MPa, when the sealing pressure was 10 MPa. This indicates that the expansion rate of IPSMP is negatively correlated with the downhole pressure. When the pressure decreases, the expansion rate becomes larger and the plugging effect gradually improves. The IPSMP expands with “memory activation” and can bridge the loss path, increase the stress around the well, and plug the formation fracture effectively, thus solving the problem of plugging the drilling tool or the failure to strengthen the wellbore due to plugging the fracture. The stress release at the same time does not cause damage to the ground. The plugging material developed by Mansour et al. can be transportable in the wellbore, and this material is capable of adapting to bridging and plugging in fractures. However, the mechanical properties of this plugging material are not effective.

Thermosetting epoxy resins not only maintain good shape-memory function in high-temperature environments, but also have better mechanical and processing properties than IPSMP [25,26,27]. Bao et al. [28] prepared an epoxy shape-memory polymer based on the reaction of anhydride crosslinking agent with epoxy polymer monomer under the action of amine catalyst. The Q20 differential scanning calorimeter (DSC) from TA, USA, can be used to test the glass transition temperature Tg of specimens with different crosslinking degrees in the temperature range of 20 to 300 °C at a ramping rate of 4 °C/min with nitrogen as the protective gas. Differential scanning calorimetry curves determine the glass transition temperature of shape-memory polymers. The glass transition of a shape-memory polymer is shown on the curve as a jump or shift in the baseline. The midpoint of the tangent line and the baseline is generally used to determine the glass transition temperature Tg (Figure 1A). As can be seen from Figure 1A, the glassy transition temperatures (72.86 °C, 88.90 °C, 102.35 °C) of the shape-recalled polymers showed an increasing trend with the increase of the cross-linking agent content. When the amount of cross-linking agent is high, the more curing cross-linking points the system forms. The tighter the curing network, the greater the inhibition of molecular chain segment movement. Therefore, a higher temperature is needed to make the material obtain the sufficient energy required for chain-segment movement, so the glass transition temperature increases. During the plugging operation, the shape-memory plugging agent is delivered to the leak layer and undergoes a heat transfer process with the formation. When the shape recovery time is not adequately controlled, two situations may occur: (1) the shape will return to the initial shape before it reaches the leaky layer, so it cannot play the role of adaptive plugging, and (2) the shape of the leak plugging agent will not return to the initial shape before it reaches the leaky layer, so it cannot play the role of adaptive plugging. The initial shape does not return in time after entering the leaky formation, so the drilling fluid cannot be bridged and plugged quickly. The leakage of drilling fluid enters deep into the formation. Therefore, it is necessary to investigate the correlation between the shape recovery rate. Moreover, the relationship between the shape recovery rate and time of specimens with different degrees of cross-linking under different temperature conditions are investigated. At a fixed activation temperature, the shape recovery rate increases with time, and the shape recovery rate is low in the initial and final stages, and high in the middle stage, exhibiting a “slow-fast-slow” trend. As the temperature increases, the shape recovery rate of the specimen increases. At Tg, Tg + 10 °C, and Tg + 20 °C, the time required to reach 90% shape-recovery rate was 2.3~18.3 min for different cross-linked shape-memory polymer systems (Figure 1B). The authors compounded it with other types of bridging plugging materials to achieve the sealing of cracks with different openings of 3~5 mm, and the sealing pressure-bearing capacity is not less than 11 MPa. Before reaching the activation temperature, this plugging agent is in the form of flakes, which can be easily transported to the leak layer and fractured with drilling fluid. After being activated by the temperature of the leak layer, it undergoes thermogenic expansion and extension, and returns to the shape of a cubic block (Figure 1C), which adapts to match the width of the fracture. However, during the experiment, they found that the epoxy polymer has its own high strength and density, so at high temperatures, it can only lead to low compression and plastic deformation. No volume change can be achieved [29,30].

In response to the above problem, Wang et al. [31] carried out studies on temperature-sensitive expandable leak plugging agent (SMP-LCM) with shape-memory epoxy resin polymer. The shape-memory epoxy foam (Figure 1D) is crushed by a crusher and passed through a standard analytical sieve to obtain temperature-sensitive expandable shape-memory plugging agent particles (SMP-LCM) in different mesh ranges (Figure 1E). SMP-LCM is a new type of intumescent plugging material based on thermosetting shape-memory epoxy resin foam. Its shape-memory characteristic is triggered by thermal stimulation, and the particle response temperature can be adjusted by controlling the ratio of internal cross-linked chain segments. The expansion rate can be adjusted by the internal pore content, and because of this, it is easy to adjust the material performance according to the leakage layer. This allows the material properties to be tailored to the application requirements depending on the leakage layer. When SMP-LCM enters the leaky channel, it expands rapidly under the effect of formation temperature, but maintains its original particle size before entering the leaky formation. This prevents the clogging of downhole drilling tools and shows a good controllability of expansion behavior. Adding hollow glass beads can effectively improve the compressibility of the shape-memory epoxy resin (Figure 1F). SMP-LCM does not require volume expansion to achieve absorption of water, which does not adversely affect the performance of working fluids such as drilling fluids. The SMP-LCM can also generate certain expansion stress to support the leaky layer pore space. It can make up for the shortage of traditional inert materials and water-absorbing resin particles. It has a good prospect for leak prevention and plugging applications [32].

To improve the toughness and mechanical properties of shape-memory epoxy polymers, Tang et al. [33] combined epoxy resins with thiols and N and N-diglycidyl-4-glycidoxyaniline (TGE) to establish cross-linked structures (Figure 1G) to obtain epoxy SMPs with sufficient mechanical strength and adequate toughness. The tensile strength increased from 32.54% to 48.36% with the increase of TGE content in the experiment. Similar to the tensile strength, the elongation of the epoxy resin specimens was 19.1%, 22.7%, 24.8%, and 26.4%, indicating that the addition of TGE increases the strength of the epoxy system, and an improvement in the epoxy material’s mechanical properties by TGE was significant, from 724.2 MPa to 1637 MPa, an increase of nearly 220%. In the experiment, TGE can fully participate in the curing process without producing other by-products, and the epoxy SMP still has good shape-memory capability [34].

Cui et al. [35] successfully synthesized shape-memory epoxy foam (SMEF) (Figure 1H) based on shape-memory epoxy polymers (SMEP) and hollow glass beads (HGBs). A new temperature-sensitive expandable plugging agent (SMP-LCM) was developed through SMEF. The plugging experiments show that SMP-LCM has good plugging effect on the high permeability loss zone simulated by the sand layer. This is mainly because the SMP-LCM can achieve its own volume expansion through the shape-memory effect (Figure 1I), which facilitates the blocking of larger loss channels by particle bridging. SMP-LCM eliminates the risk of downhole drilling tool plugging and enhances the plugging effect compared to conventional efficient plugging materials LCM (e.g., screw shell or fiber optic).

Wang et al. prepared a series of thermosensitive polymer nanospheres with different low critical solubility temperatures (LCST), named SD-SEAL (Figure 1J) [36]. The experimental results show that the synthesized nanoparticles are temperature sensitive and have a significant LCST value that increases with increasing hydrophilic monomer AA. When the temperature is higher than its LCST value, SD-SEAL acts as both a physical seal and a chemical inhibitor, slowing down the pressure transfer and significantly reducing the shale permeability (Figure 1K). The change of the shale blockage layer to a hydrophobic layer has greatly improved the stability of the shale.

**Figure 1 molecules-28-02975-f001:**
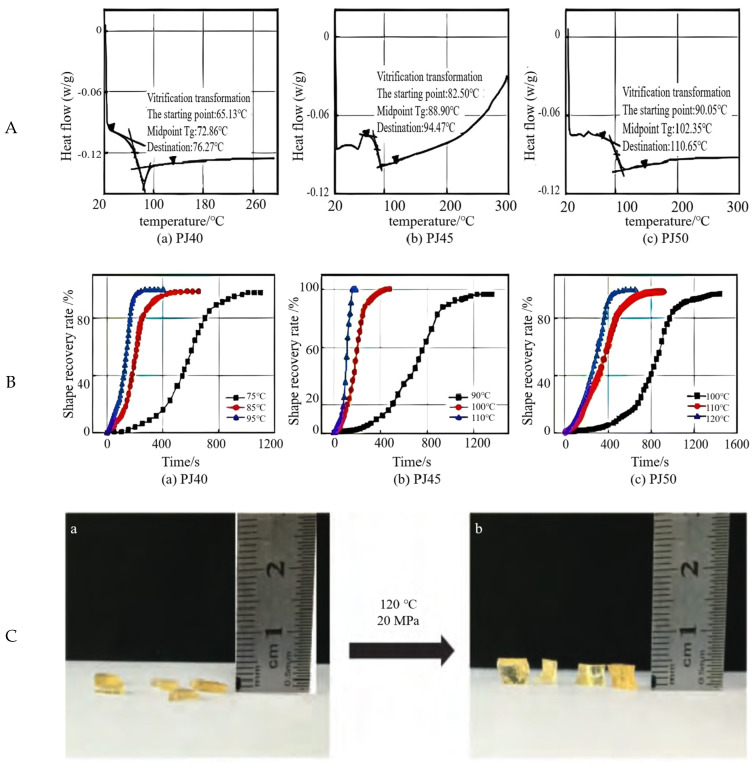
(**A**) Differential scanning calorimetric analysis curves of shape-memory polymers with different degrees of cross-linking [28]. (**B**) Shape recovery rate versus time for shape-memory polymers under different temperature conditions [28]. (**C**) Shape-memory plugging agent before and after activation under high temperature and pressure conditions. (**a**): Activated before; (**b**): After activation [28]. (**D**) Shape-memory epoxy resin synthesis reaction [31]. (**E**) SMP-LCM in different mesh ranges [31]. (**F**) Shape-memory epoxy foam expansion rate versus the temperature of different hollow glass bead dosage (curing agent dosage is 17%) [31]. (**G**) Cross-linking structure diagram [33]. (**H**) Synthesis of shape-memory epoxy foam [35]. (**I**) Status of the SMP-LCM in the depletion channel before and after the extension [35]. (**J**) Synthesis route of SD-SEAL nanoparticles [36]. (**K**) Diagram of SD-SEAL physical blocking and chemical inhibition [36].

IPSMP does not have a shape-memory effect, but possesses a shape-memory function after special processing [37]. The mechanism of action is: when the temperature is lower than the glassing temperature, the IPSMP particle size is small and subsequently enters the leakage layer with drilling fluid, forming high-density particles that fill the channel; and when the temperature rises to the glassing temperature, IPSMP “activates” and undergoes expansion and deformation, and the high-density particles squeeze and fill each other, forming a high-strength bridging accumulation, which can effectively seal the formation cracks and strengthen the wellbore [38]. The resulting stress inflicts no harm to the formation and does not influence the normal work of downhole machinery [39]. It has the properties of low density, adaptive bridging and plugging, adjustable activation temperature and time, etc. [40,41]. It has important practical value and development potential for the treating of well leaks in complex formations (fractured leaky formations) and it is expected to achieve technological innovation in leak prevention and plugging [42]. Although IPSMP has a broad application prospect in drilling and plugging, most of them are still in the laboratory stage, and there are not many field application studies. Therefore, the author suggests that there is still a need to explore improving the phase change reliability, ductility, and chemical durability.

## 3. Intelligent Polymer Gel

Intelligent polymer gels are subject to intelligent changes in the external environment, and the hydrophobic interaction, hydrophilic interaction, van der Waals forces, and electrostatic forces between ions in the gel polymer compete with each other to cause conformational changes in the polymer chain segments, which eventually lead to phase changes. Gel is one of the commonly used materials for leak prevention and plugging, and there are two processes of this form of plugging: gel plugging and gel plugging with drilling when used on site. Gel plugging is to inject the gel solution into the leakage channel for plugging. The characteristics of this process are that the gel fills the leaky channel to a high degree and has a continuous gel formation, which results in a strong pressure-bearing capacity; however, the disadvantage is that the plugging period is long. Gel plugging with drilling is to disperse the gel particles in the drilling fluid, forming a plug by bridging in the leaky channel; the advantage is that the plugging with drilling is easy to operate, and the disadvantage is that the buildup layer is easily washed away. Intelligent gel plugging process needs to combine the advantages of the above two processes, so as to not only achieve plugging with drilling, but also to realize the formation of the continuous gel in the leak channel to seal the strength.

At present, most intelligent polymer shape-memory polymers have many limitations to the malignant loss layer, such as large cracks, large pores, high temperature, pH and other harsh conditions [43]. In order to improve the success rate of gel plugging in the malignant loss layer, the gel must have excellent temperature resistance, salt resistance, expansion, gellability, pressure resistance, and other properties [44]. Therefore, intelligent polymer gels (IPG), a multifunctional system consisting a low-molecular medium and a polymer with a three-dimensional cross-linked network structure, was created [45]. IPG takes natural macromolecular cellulose as the backbone. By adding a cross-linking agent and catalyst, it reacts with copolymerized water-soluble acrylamide gel and reasonably controls the gel-forming time to achieve the purpose of construction injection [46]. After entering the ground, IPG is “awakened” under certain temperatures and pressure, and molecules interact with each other, resulting in accumulation, retention, and bridging. The cracks can be filled and reinforced to improve the compressive strength in the leaky layer and successfully seal the leaky layer [47]. Herein, the author outlines the advantages and disadvantages of three common IPG systems and commonly used products (Table 1), and elaborates on four aspects: temperature resistance, salt resistance, pressure-bearing, and toughness.

### 3.1. Temperature Resistance

In recent years, the number of ultra-deep wells has increased, and the plugging agent must have excellent temperature resistance [63,64,65]. The molecular chain of a cross-linked polymer gel plugging agents is easy to break under high-temperature conditions, resulting in changes in the nature of the sealing tape, which destroys the structure and hence fails to achieve the purpose of sealing [66]. To solve such problems, researchers have conducted experimental studies by introducing anti-temperature groups, molecular structure design, and adding nanomaterials or cross-linking agents.

Li. [67] used sodium styrene sulfonate (SSS) and 2-Acrylamide-2-methylpropanesulfonic acid (AMPS) as raw materials, by introducing sulfonic acid group (-HSO3) to improve the temperature resistance of the product, and developed high-temperature-resistant water absorption resin sealant. The expansion properties show that the gel has good expansion and the expansion rate becomes more extensive with the increase in temperature (Figure 2A). After heating at 180 °C for 12 h, the expansion rate can reach 2.5 times. If the heating is continued, the expansion rate of the gel will remain at approximately the same value. This indicates that the gel is able to maintain good expansion under high temperature conditions. The gel can be formed after heating at 180 °C for 2.5 h. After 20 days of continuous heating, the gel still has good stability. This is because the network structure of the plugging agent becomes denser with the addition of AMPS. After introducing -HSO3 into the molecular chain, the two groups (AMPS, -HSO3) cooperate with the temperature resistance, so the thermal stability of the plugging agent is significantly improved.

Through molecular structure design, Zhang et al. [68] used acrylamide (AAM), butyl methacrylamide and AMPS as monomers, flexible fibers as reinforcing materials, ammonium persulfate (APS) as an exciter, and reacted them with a homemade macromolecular bridging agent BWL to make a high-temperature-resistant fiber reinforced gel plugging agent. Microgel BWL is used as a cross-linking agent, which is firstly a reactive monomer, capable of its own polymerization and controlled polymerization conditions to obtain a partially polymerized microgel. Partially polymerized microgels are produced by polymer macromolecular chains with reactive termini (amide groups, carboxyl groups, acyl groups, etc.), then cross-linking them with organic carboxyl and acyl groups, etc., and finally reacting them with organic cross-linking agents. The reaction with organic cross-linking agents results in microgels containing a large number of reactive C=C double bond groups. Since BWL contains a large number of C, the crosslinking density of BWL with monomers is much higher than that of monomers with conventional organic crosslinkers, thus forming a dense three-dimensional mesh structure with good thermal stability. The stable value of the energy storage modulus of the gel is about 5500 Pa before the addition of flexible fibers. The energy storage increases significantly to about 7200 Pa after the addition of flexible fibers (Figure 2B). The blocking rate reaches 98% at 100 °C (Figure 2C). When the temperature increases to 250 °C, it still has good stability. The mass retention of IPG remained high at 350 °C, which indicates that the plugging agent has good temperature resistance. The author believes that the addition of flexible fibers can physically cross-link with the main chain and side chains of the gel, making the three-dimensional network structure of the gel more compact, which in turn increases the toughness of the gel and enhances the temperature resistance of the gel.

He et al. [69] developed DE-modified polyacrylamide hydrogels by a cross-linking reaction using diatomaceous earth (DE) nanoparticles as reinforcing raw materials. From the analysis graphs of temperature resistance before and after the addition of DE (Figure 2D) and an analysis of the graph of the glassy temperature (Figure 2E), it can be seen that the degradation temperature of the gel polymer backbone is increased to 517.0 °C at a DE concentration of 0.09% (mass fraction); while the glassy temperature (Tg) is increased to 135.2 °C. This indicates that DE nanoparticles function as a thermal insulation and mass transfer barrier, preventing the gel from thermal degradation due to high temperature. The DE nanoparticles form hydrogen bonds with the amide and carboxylic acid groups of hydrolyzed polyacrylamide (HPAM) through the hydroxyl groups on the silica surface (Figure 2F). Then, it embeds in the gel and interacts with the polymer chains, increasing the resistance to molecular motion, which leads to an increase in Tg. Therefore, the thermal stability of IPG is significantly improved.

Qu et al. [70] developed a new acrylamide copolymer/phenolic resin gel with phenolic resin as the cross-linking agent. After cross-linking, the gel completely loses its fluidity and becomes a high-strength plugging agent with temperature resistance of up to 200 °C. Through the sand bag test, the permeability test of the gel-treated core and the comparison test of other plugging materials, it is shown that the gel has good plugging performance. The gel can be mixed with other solid plugging agents to improve plugging performance without affecting its performance. It has a good development space in well loss control, workover operations, water plugging, and other aspects.

Liu et al. [71] established a gel system formed by terpolymer (L-1) and a new cross-linking system (HB-1) to study its thermal stability at ultra-high temperatures. The experiments showed that the gel system could form a stable continuous three-dimensional network structure at 120–200 °C, which indicates that the gel system has good long-term thermal stability. Compared to conventional high-temperature-resistant gels, the cross-linked system HB-1 contains four hydroxyl groups (-CH2OH). Cross-linking with four amide groups (-CONH2) on the terpolymer can form a three-dimensional network structure with a smaller mesh size (less than 5 μm), thereby strengthening the thermal stability. At 200 °C, the grid size of the network structure is about 10μm, but the strength of the gel system can still be maintained at the F-G level. Through scanned electron microscope (SEM) images, the gel system is shown to be very firmly attached to the porous media, which can realize the adjustment of the formation profile. The author believes that the gel system has great potential in the application of high temperature oil and gas reservoirs.

### 3.2. Salt Tolerance

When the mineralization degree (MD) of formation water is too high, the stability of polymer chemical bonds will be destroyed [72]. This leads to changes in the spatial structure of the polymer gel, causing changes in its physicochemical properties and thus losing the sealing effect on the leaky layer [73]. The water absorption performance of IPG is affected by high salinity. The high mineralization makes the IPG expansion multiplier lower, and the result is that the polymer gel is unable to seal wider cracks [74].

Because single polyacrylamide in high temperature environment will affect the dissection and plugging performance, it causes a reduction in the cross-link strength of the gel. To enhance the temperature and salt resistance of cross-linked polymer gels, Zhu. [75] tested the addition of partially hydrolyzed polyacrylonitrile (HPAN) and modified silica to partially hydrolyzed polypropylene amine(HPAM). As seen in Table 2, the gelatinization time is 48 h at 90 °C. At 120 °C and MD of 20 × 10^4^ mg/L, the gel strength remains almost unchanged for 90 days with no sign of broken gel hydration, and it indicates it has excellent temperature and salt resistance. This is because the modified ultrafine SiO_2_ has better dispersion, and it can serve as a stabilizer for HPAM and HPAN. Due to the cross-linking of these two polymers, the temperature and salt resistance of the gel have been improved. This plugging agent has achieved a good effect of precipitation and oil increase in the Qingzu oil production area of Huqing Oilfield.

Wang et al. [76] developed a temperature-resistant and salt-resistant page. In it, the authors detail the following step-by-step guide: allow AAM, AMPS, methacrylic amide and propyl trimethyl ammonium chloride (DTAP) to dissolve completely in water and then adjust the pH value. After deaerating, add the initiator and other closed reactions to form colloidal particles. At 90 °C, add granular alkali for hydrolysis for 6 h before adding a thermal oxygen stabilizer. Finally, after drying, crush and screen the contents to produce the product called BHGL. From the effect of different temperatures and the mineralization on viscosity and viscosity stability curves (Figure 3A), it was found that at 85 °C, the salt resistance reaches 32,000 ppm. Compared with PAM, the viscosity, thermal stability, and salt resistance are significantly improved. The analysis shows that the introduction of AMPS in the gel makes the linear polymer more hydrophilic and improves the salt resistance of the molecular mechanism.

In addition, Qin Yi et al. [77] developed a nano-composite gel (NCPG) by adding a salt-resistant sulfonic acid group (-HSO3) and strong adsorption nanoparticles into the gel particles. This composite gel is stored for 48 h at 80 °C in formation water with an MD of 81,521.2 mg/L. Its water absorption expansion rate can reach 12.4 times and remain stable for more than 30 days, which is 33.7% lower than that of water with the same temperature (Figure 3B). This shows that the water absorption rate of NCPG is controllable. Introducing -HSO3 into the gel particles can improve the salt resistance of NCPG. Laponite itself has excellent dilatancy, which makes NCPG have excellent dilatancy in a high MD environment.

Wang et al. [78] introduced a salt-tolerant monomer AMPS through an aqueous solution. The absorbent resin gel of acrylic acid-2-acrylamido-2-methylpropanesulfonic acid copolymer [P(AA-AMP)] is developed by using tetraallyl ammonium chloride (TAAC) as a bridging agent and potassium persulfate (KPS) as an initiator. The changes in water absorption properties of water absorbent resin in different salt solutions were studied experimentally at 150 °C (Figure 3C). Its water absorption rate can reach 78 g/g in 1% NaCl. This shows that the water absorbent resin gel has excellent temperature and salt resistance. The prepared water-absorbent resin has good high temperature resistance and is stable in 100 to 200 °C. The water absorption performance is stable within 200 °C. The absorbent resin has excellent acid and alkali resistance in a salt solution, and can maintain relatively high absorption multiplicity between pH = 5~9. In different types of Cl^−^ salt solutions, the absorbency of the absorbent resin decreases with increasing concentration of the salt solution. The higher the cationic valence at the same concentration, the lower the absorbency. The water-absorbing resin gel has achieved good results in experiments, expanding its use within the high-temperature and high-salt petroleum industry.

Yang et al. [79] developed an amphiphilic polymer gel by cross-linking a salt-tolerant amphiphilic polymer with organ chromium. At high salinity (NaCl 80,000 mg/L) and temperature (85 °C), the gel polymer shows a very low synergistic effect with organic chromium. Bottle test method and breakthrough vacuum method are used to determine the gelling time and gelling strength, and the plugging behavior is analyzed by sand-covered pipe experiment. The results show that the gels formulated 3000 mg/L polymers with 0.3% (mass fraction) of a bridging agent. The gelation time is 4 h, the gelation strength is 0.037 MPa, the stability of the body gel is 93%, and it can be stable for about 60 days and has a good sealing effect. This suggests that amphiphilic polymer gels with low polymer concentrations are more effective in plugging the high-salinity loss of circulation incidents.

### 3.3. Pressure-Bearing

In the process of petroleum exploration, with the increase of formation depth, not only does the temperature and salinity increase, but the pressure also increases [80]. Therefore, it is also necessary to improve the pressure-bearing performance of IPG to achieve leak plugging. 

Chen et al. [81] prepared a high acid soluble fiber leakage plugging agent (SDSF). It is combined with other types of high acid soluble bridging plugging materials and it can adjust the working fluid formulation for different opening wedge gaps. Adding 0.3% SDSF near the crack opening end of the plugging bridge can improve the structure of the crack plugging layer. It is found to have a pressure-bearing capacity of 10 MPa and a leakage volume at only 50 mL. The main sealing mechanism of SDSF can be summarized as follows: ① fiber bending and squeezing friction, which means that the form of fiber in the crack sealing layer is bending and turning. When the sealing layer is subjected to an external force to bend the fiber under tension, the concave side of the fiber bending will generate pressure and friction on the granular plugging material, preventing the displacement of the granular plugging material and enhancing the compressive strength of the sealing layer, as well as improving the pressure-bearing capacity of the sealing layer; and ② fiber interwoven into a network of three-dimensional tension (that is, there are many intersections between the uniformly distributed fibers in the crack plugging layer) when the fibers are displaced by external forces, other fibers will prevent this displacement. The deformation of any section of fibers will move the fibers in all directions, forming a three-dimensional force zone, which improves the shear strength of the blocking layer and enhances the stability of the blocking layer.

Fang et al. [82] prepared a polymer gel (HND-1) by combining various nano-scale plugging raw materials, such as the retarder HN-1 and plugging agent DL-1. HND-1 can be thickened within 4~20 h and has a strong retention ability. The compressive strength of HND-1 can reach more than 12 MPa within 24 h. Its operating principle is displayed in Figure 4A: HN-1 molecules can buffer the crosslinking reaction and form a dense network structure, which plays a bridging role; the formation is further blocked by the reaction of small molecule gelling agent and cross-linking agent, and the inorganic granular plugging agent fills the small gap, forming a “multi-element collaborative plugging system”. It can greatly improve the pressure performance of fractured and cave-type formations, and the compressive strength of the gel is more than 10 MPa within 20 h of curing. The author believes that HND-1 gel is especially suitable for the pressure sealing of long open-hole formation.

The low pressure-bearing capacity of the well wall in fractured formations will cause a leak of drilling fluid [83,84]. Wang et al. [85] used a plugging agent to plug the leakage cracks to solve this problem. The load-bearing capacity of the rock formation is increased by forming a dense sealant film on the surface of the rock wall (the principle is shown in Figure 4B) and by combining the leakage characteristics of the fractured strata in the LX block. According to the principle of multi-synergy, based on parallel and the trapezoidal fracture permeability physical model, the optimization of plugging material, suspension stability of confined slurry, and the confined plugging performance were all tested by traditional plugging instruments and new experimental devices, respectively. The author has thus developed a set of acid-dissolvable plugging formulations of pressure-bearing plugging agents suitable for conventional formations and reservoirs. Laboratory evaluation results show that the developed plugging agent has an excellent anti-pressure plugging function, and the effective pressure value is as high as 28.0 MPa. It can effectively seal downhole hydraulic fractures smaller than 2.0 mm and satisfy the technical requirements of downhole completion pressure plugs.

Zhang et al. [86] developed a self-breaking supramolecular plugging system for controlling leakage from workover operations that overcomes the drawbacks mentioned above with regard to conventional LCM. The system forms a spatial lattice structure through intermolecular non-covalent bonds with carbon, oxygen, and sulfur bonds. After the gel has cured, the supramolecular sealer is a semi-solid gel with high viscoelasticity and strength. The supramolecular gel effectively seals the sand contact surface. The gel has a sand bed bearing capacity of 4.5 MPa at 120 °C. It can effectively plug low pressure and leaky formations at 90–120 °C. The technical ideas and innovations of self-breaking supramolecular plugging agents are important references and demonstrations for the exploration and development of low pressure and low permeability reservoirs around the world.

In fractured formations, conventional plugging materials are of an inappropriate size and low strength, resulting in an unsuccessful initial plugging and increased production costs [87,88]. For this reason, Fan et al. [89] developed a polymer plugging gel for fractured formation (XNGJ-3, as shown in Figure 4C). XNGJ-3, a reactive monomer mainly consisting acrylamide monomers with carboxyl and hydroxyl groups, has low viscosity before gelling, and it becomes gelled at 80 °C, which is conducive to the smooth entry of plugging materials into the cross cracks and sealing of cracks. It has good deformation ability and can prevent damage in the process of crack closure. The strength of XNGJ-3 is improved by adding fiber bridge material. The simulation results of the lost well show that the bearing capacity of the material can reach 21 MPa, and the reverse bearing capacity can reach 20 MPa. The author’s investigation and analysis find that the pressure-bearing strength of XNGJ-3 is more than twice that of ordinary polymer blocking gel.

**Figure 4 molecules-28-02975-f004:**
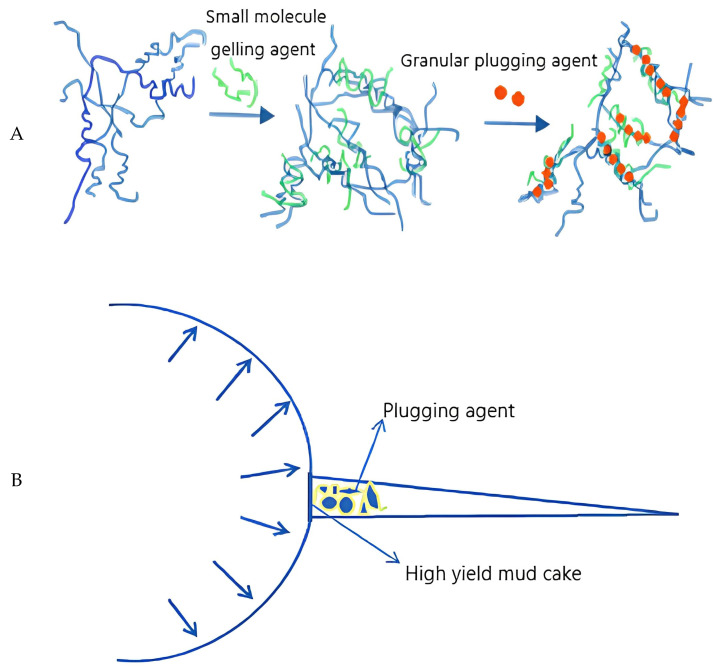
(**A**) “Multi-far Cooperative Blocking” schematic diagram [82]. (**B**) Principle of improving crack compressibility [85]. (**C**) (**a**): Physical diagram. (**b**): molecular structure formula [89].

### 3.4. Toughness

Under the pressure of formation water and wellbore working fluid, the plugging zone formed by IPG may suffer shear failure [90]. Therefore, the sealing belt needs to have a certain toughness, otherwise it will cause the rupture of the sealing belt, and the sealing cannot be realized.

Liu et al. [91] added Fe^3+^ to the gel network structure of polyacrylamide. A hydrogel system was prepared by adding AA monomer and introducing carboxylic acid root for coordination. The coordination ratio of both is 1:3. The coordination of cations can quickly combine and disperse, and it is easy to dissipate energy and recover mechanical properties. Therefore, the toughness of the gel is improved, the brittle strength of the gel is up to 5.9 MPa, and the elongation is about eight times. 

Chen et al. [92] prepared ductile polyacrylamide hydrophobic cross-linked gels (HPAAm) by the gradient-induced cross-linking of micelles of hydrophobic monomers, stearoyl acrylate (STA), and acrylamide. HPAAm remains intact after compression to 0.9, while the corresponding polyacrylamide gels prepared by adding chemical cross-linking agents have been destroyed. Meanwhile, the elongation at the break of HPAAm can reach 1000%, which is also much higher than the conventional polyacrylamide gel. It indicates that HPAAm gel has high toughness. In addition, by adding another polyacrylamide (PAAm) to HPAAm, a double-network polyacrylamide hydrogel (DN-HPAAM) containing hydrophobic cross-linking can be formed. The synthesized DN-HPAAM has better toughness strength (up to 2 Mpa) and a better cyclic compression performance than HPAAm.

In addition, Cui et al. [93] prepared a high toughness hydrogel PA based on polyamphoteric electrolyte. PA gels were prepared by a one-step method using free radical polymerization. Firstly, an aqueous solution was prepared containing anionic monomers, cationic monomers, and free radical initiators. Then, the precursor gel was obtained by UV-initiated polymerization under argon atmosphere (as-prepared gel). Finally, the precursor gel was gelled into a large amount of water for dialysis to obtain the equilibrium gel. During dialysis, the salt ions in the precursor gel enter the surrounding water, thus promoting the formation of ionic bonds by combining positive and negative ions within and between polymer chains. It is clear that PA gels and double chemical network hydrogels have similar toughening mechanisms (Figure 5), although their topologies are very different. Strong bonds form the network backbone to maintain the gel shape, while weak bonds act as sacrificial bonds to dissipate energy and toughen the gel. Zhao Dan et al. introduced epichlorohydrin after adding cellulose to the gel [94]. This makes cellulose molecules bond through covalent bonds to form chemical crosslinking, and it makes an ample slip space between molecules, thus increasing the toughness of the gel. However, these studies are only preliminary. It should be combined with multi-scale research tools to systematically study its structural changes and damage laws at various scales under different deformation modes (such as creep, stress relaxation, fatigue, etc.), which is essential to understand its damage behavior and reveal its damage mechanism.

## 4. Intelligent Polymer Membrane

According to the film-forming property of polymer emulsion, it is added to the drilling well fluid, so that the polymer particles are uniformly dispersed in the drilling fluid. As the drilling fluid is filtered out into the formation, the polymer particles and solid phase particles settle on the well wall. When the drilling fluid leaches out into the formation, polymer particles and solid phase particles settle on the well wall. The polymer is gradually confined to the pore space of the mud cake. With further filtration loss, the amount of water in the pores decreases gradually. The polymer particles flocculate together, forming a polymer seal layer on the surface of the solid phase particles. It binds the solid phase particles and the surface of the well wall, and the pores in the mud cake are filled with cohesive polymer. Due to the continuous filtration loss, the water between the cohesive polymer particles is gradually lost. Finally, they are completely coagulated together to form a continuous polymer mesh film structure. The solid-phase particles in the mud cake are intertwined with the polymer, improving the quality of the mud cake and preventing drilling fluid from entering the formation. This increases the bonding strength of the mud cake to the formation, which plays a role in stabilizing the well wall.

Geological drilling and oil-gas development is gradually developing in depth. In the process of drilling, there are problems such as collapse, leakage, and reservoir pollution caused by borehole wall instability [95,96,97,98]. Drilling fluid film forming technology (adding film forming material to drilling fluid can generate permeable film or diaphragm on the surface of the plugging layer, preventing or reducing drilling fluid from entering the formation, preventing the formation from producing cracks, and maintaining wellbore stability) can effectively solve such problems, and so this technology has become a hotspot for research [98,99,100].

Song et al. [101] researched the film formation mechanism of styrene butadiene rubber (SBR) polymer emulsions. After the emulsion polymers evaporate through water, the polymer particles approach each other and combine to generate a blocking film (Figure 6A). A visible sand bed of 20 mm thickness was prepared by using quartz sand of 0.18–0.28 mm grain size to simulate a high permeability funnel formation. The corresponding pore throat size was 0.056 mm, and the loss of drilling fluid filtrate and the depth of filtrate intrusion were tested by means of pressure measurements. The experimental conditions were 25 °C, 0.7 MPa, and 7.5 min. The drilling fluid formulation was 3% bentonite slurry + 3% bridging filler particles + 4% ultrafine calcium carbonate + 5% SBR polymer. A 30 μm microfracture was simulated with a self-modified microfracture plugging evaluation device, and the drilling fluid was taken from the site. First, they evaluated the effect of 5% SBR polymer addition on the performance of drilling fluid. According to the 2/3 bridging theory, 3% of calcium carbonate with a particle size of 0.02 mm was added to the drilling fluid to conduct high-temperature and high-pressure filtration loss experiments. Through the above polymer emulsion sealing effect experiments, SBR polymer can form a dense mesh-like film structure in the mud cake and form irregular fibrous bridge plugs on the surface of cracks and pores. It can act synergistically with the bridging particles to improve the bridging plugging efficiency and reduce filtration loss. The application of SBR film-forming fluid can form a mesh-like reinforced layer on the well wall, improve the dense and flexible filter cake, reduce the percolation effect of drilling fluid, and be a good solution to the problem of well wall instability in deeply fractured formations, providing technical support for oilfield deep drilling and development.

Zhu et al. [102] introduced the warm pressure film forming agent (PF-HCM) to deal with the borehole wall instability caused by filtrate intrusion into reservoir during drilling in Hade Oilfield. After contact with water, it becomes “soft on the outside and rigid on the inside”, and forms a film with a certain strength through polymerization, fusion, and cross-linking (Figure 6B). It can resist high temperature, resist the erosion of drilling fluid, and inhibit free water in the slurry from penetrating into the well wall. Under the aging condition of 120 min filtration losses of 173 mL and 110 °C × 16 h, and adding 2%PF-HCM, the filtration loss is only 8.3 mL. Experimental results show that adding 2%PF-HCM to the PEM drilling fluid system can significantly reduce the loss and filter cake permeability of the system without affecting the rheological properties of the system. This plugging agent shows good plugging and collapse prevention ability in three permeability levels: low, medium, and high in the core. In order to popularize and apply the new PF-HCM, it is necessary to study further and improve the plugging and collapse prevention technology.

In addition, in relation to the characteristics of Ordos Basin, Wang et al. [103] proposed the idea of “enhancing the sealing and anti-collapse property, and complementing inhibitive and lubricity”, and prepared a nano-film sealing agent. Under the test conditions of 100 °C and 4.2 MPa pressure, the leakage volume does not exceed 2 mL. Data show that the seam range of mud shale reservoirs in China is between 5 and 200 nm. The particle size of the developed nano-film-forming plugging drilling fluid is 148 mm. It has significant advantages in sealing tiny gaps. The plugging agent developed by this team has a strong plugging and inhibiting effect. When the specific gravity is 2.2 g·cm^3^, it can resist temperatures of up to 150 °C, and has an obvious stabilizing effect on the well wall. The plugging agent has only been used for four wells in production and exploration wells, and should be developed and popularized in the later stage.

When the intelligent polymer film plugging material enters the formation, it is “activated” to bond with the wall around the crack under the influence of the formation environment, forming a film layer with high thickness and toughness. They can realize the “shrinkage” of the crack until it is blocked (Figure 6C) [38]. Intelligent membrane has good filtration loss and plugging effect, well stabilization, and cementing function. It does not have toxicity and causes no pollution to the environment, but its research and development costs are high, and plugging and collapsing prevention technologies are not mature.

## 5. Intelligent Polymer Bionic Plugging Material

In the 21st century, a bionic mindset is emerging in all areas of engineering [104]. In terms of petroleum engineering, bionics theory is not only used to deal with problems such as drilling and pipeline protection, but also has a significant impact on the thinking concept of technological innovation in the petroleum industry [105]. This subsection presents some important results of bionics in dealing with well wall stabilization, collapse prevention, and plugging problems.

The bionic drilling fluid system is a new drilling fluid technology used to stabilize the well wall. This technology is based on the phenomenon of marine mussel foot filament protein, which has a strong adhesion ability to adhere and encapsulate on deepwater rocks. The researchers have developed a water-soluble polymer with similar characteristics as a drilling fluid additive, thus forming a new bionic drilling fluid system. When the bionic drilling fluid system comes into contact with the well wall rock, the bionic treatment agent in the system can be spontaneously solidified by “spotting” to form a strong high strength adhesive gel “bionic shell”. It can improve the strength of the near-well wall rock and seal the rock’s microporosity. Through the synergistic effect of “strengthening, blocking and inhibiting” to achieve the unification of internal and external factors, it can effectively solve the problem of wall instability.

Jiang et al. [106] conducted research on bionic solid wall drilling fluid. The bionic drilling fluid system is based on the bionic cementing agent GBFS-1. The formulation of the bionic drilling fluid system is as follows: 4% sodium-based bentonite + 0.3% Na2CO3 + 0.2% NaOH + 0.6% amphoteric polymeric filter loss reducing agent + 2% sulfonated asphalt FT-1A + 2% emulsified asphalt FD-1 + 2% bionic wall-fixing agent GBFS-1 + 1% liquid lubricant + barite (density of 1.20 to 1.30 g/cm^3^). The results show that the hot rolling recovery of shale is 90.6% and the compressive strength is increased by 10.8% under the influence of bionic cemented mud. By analyzing the chemical composition of mussel foot silk protein (MFSP), we know that it has a functional group. It can be grafted onto a polyol framework to synthesize bionic cementing agent GBFS-1. Its chemical structure and properties are similar to MFSP. It has strong adhesion under water. It can spontaneously cross-link and solidify on the surface of mud shale and become a “bionic shell” with strong cohesion and adhesion, which improves the strength of the bionic solid wall in the drilling fluid system, prevents hydration and evacuation of mud shale, and strengthens the well wall.

On this basis, Cheng et al. [107] introduced the bionic treatment agent on the basis of the original compound salt formula to solve the problems of well collapse, well loss, and diameter reduction in well 3-21X, which can improve the stability of the well wall. By adding sodium chloride to obtain a bionic solid complex salt drilling fluid system. The results show that the system has good inhibition. The recovery rates of salt rock and mudstone are 85% and 95%, respectively. The rheological properties remain unchanged when 0.6% calcium chloride is added, indicating that its comprehensive properties are easy to be regulated. It can reduce the well diameter expansion rate by an average of 62%, thus achieving the effect of stabilizing the well wall. This drilling fluid shows good inhibition in solving the problems of hydration, expansion, and scattering of silt rock, salt-paste rock and paste mudstone, and can effectively stabilize the borehole wall and prevent collapse.

In recent years, Liu Wei et al. developed a borehole solidified bionic drilling fluid system in order to solve the problem of borehole instability in Moxi, Sichuan [108]. In the bionic drilling fluid system, the inhibitor CQ-YZF is embedded in the clay interlayer domain by ion exchange, and binds the adjacent clay crystalline layers together by strong hydrogen bonding, which greatly inhibits the hydration of clay. This can greatly inhibit the hydration and swelling of clay, and play a “microscopic” role in wall consolidation. The solidifying agent CQGBF spontaneously adsorbs on the near surface of mud shale and cross-links with Ca^2+^, Mg^2+^, and other metal ions on the surface of mud rock through “bionic groups”, and solidifies to form a gel film with strong adhesion and cohesion. The “bionic shell” can improve the cementation strength of the mud shale and strengthen the mud shale. Experimental results show that the plugging rate of the bionic plugging agent is more than 90%. It shows that the drilling fluid system has a good plugging performance of micropores. As the drilling fluid pressure increases, the plugging agent CQ-NWD generates ultra-thin and compact filter cakes on the surface of the lost rock and spreads them evenly, achieving good results in plugging nanomicron cracks.

To sum up, the intelligent polymer bionic plugging material contacts with the well wall rock after entering the formation, and automatically solidifies into a high strength adhesive gel “bionic shell” by using its biological super adhesion ability. It can effectively improve the strength of rock strata near the shaft wall and plug the micropores. It has an excellent performance in high permeability and microfracture leakage formation. Moreover, it has an excellent inhibitory effect on the hydration, expansion, and dispersion of salt-gypsum rock and gypsum mudstone, and can stabilize and strengthen borehole wall and prevent collapse effectively. Intelligent polymer bionic plugging material has a low filtration loss and good thermal stability, and it can reduce the risk of well leakage. However, the research technology of this material is not mature, and it will take some time for it to be widely used.

## 6. For Reviews and Perspectives

The intelligent plugging material can intelligently adapt to various complex formations and has excellent mechanical properties. It can significantly improve the plugging efficiency and has a broad application prospect in the field of drilling fluid leak prevention and plugging. Intelligent shape-memory plugging material has the advantages of strong pressure-bearing capacity, adaptive bridging plugging, adjustable activation temperature, etc., which can be applied to fractured leaky strata. Intelligent gel plugging material has the advantages of strong self-adaptive ability, good compatibility, strong scouring resistance and good degradability, etc. It can be applied to seal fracture and cavity leakage strata. Intelligent membrane and intelligent bionic plugging materials have the advantages of good plugging performance with drilling and solving co-existing technical problems such as leakage jamming, collapse, and formation damage, etc. They are suitable for high permeability and microfracture leakage formations. The combination of intelligent materials and drilling fluid plugging technology can be divided into three stages: basic research, the transformation of results, and industrial application. In future research, intelligent shape-memory materials should focus on improving their compressive strength through a two-stage curing technology. Intelligent gel materials should improve the applicability of self-healing, self-adhesiveness, and other with gel materials under harsh conditions, such as high temperatures. Intelligent membrane plugging materials should realize the double enhancement of film thickness and film strength, and improve the applicability of intelligent membrane to leaky channels. Intelligent bionic materials should focus on the combination of bridging blocking, adaptive blocking, and intelligent bionic blocking to achieve synergistic reinforcement of leak plugging. 

In general, the research and application of intelligent materials in the field of drilling fluid plugging are still in the initial stage, and the performance of intelligent plugging materials should continue to be improved. The scope of use should be broadened in the future. We should develop a scientific and intelligent plugging process, accelerate the transformation of results and industrial application, and promote the practical, innovative, and intelligent development of drilling fluid leak prevention and plugging technology.

## Figures and Tables

**Figure 2 molecules-28-02975-f002:**
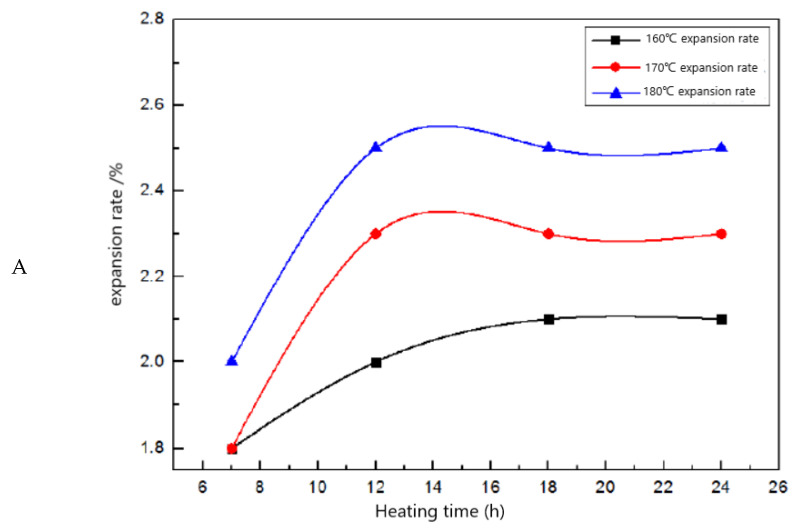
(**A**) Anti-high temperature plugging gel expansion rate [67]. (**B**) Curve diagram of the influence of flexible fiber content on the stable value of storage modulus. (**a**): Before adding flexible fibers. (**b**): After adding the flexible fiber [68]. (**C**) Thermogravimetric analysis curves of high-temperature-resistant fiber reinforced gel particles [68]. (**D**) Analysis of the temperature resistance before and after adding DE. (**a**): DE-0; (**b**): DE-9 [69]. (**E**) Analysis of the glassing temperature before and after adding DE. (**a**): DE-9; (**b**): DE-0 [69]. (**F**) Structure diagram of DE and HPAM forming hydrogen bond [69].

**Figure 3 molecules-28-02975-f003:**
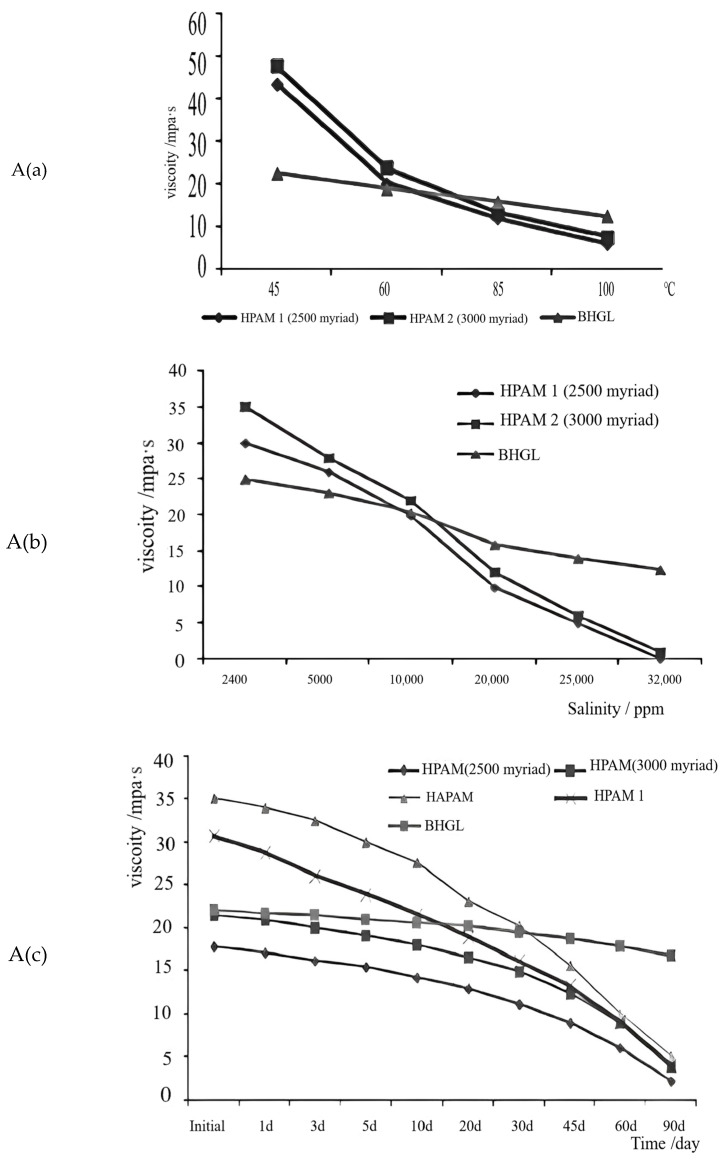
(**A**): (**a**) Trend graph of viscosity at different temperatures, (**b**) effect of different mineralization degrees on viscosity, (**c**) viscosity stability curve [76]. (**B**): (**a**) Expansion multiplier curve in ground water, (**b**) expansion multiplier curve in clear water [77]. (**C**) Absorption multiplicity curves in different salt solutions [78].

**Figure 5 molecules-28-02975-f005:**
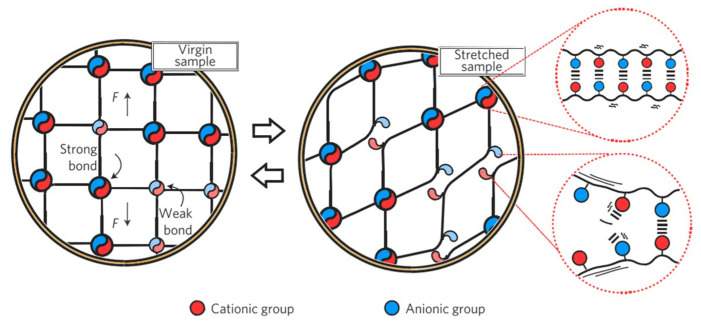
Schematic diagram of PA gel network containing ionic bonds of different strengths [93].

**Figure 6 molecules-28-02975-f006:**
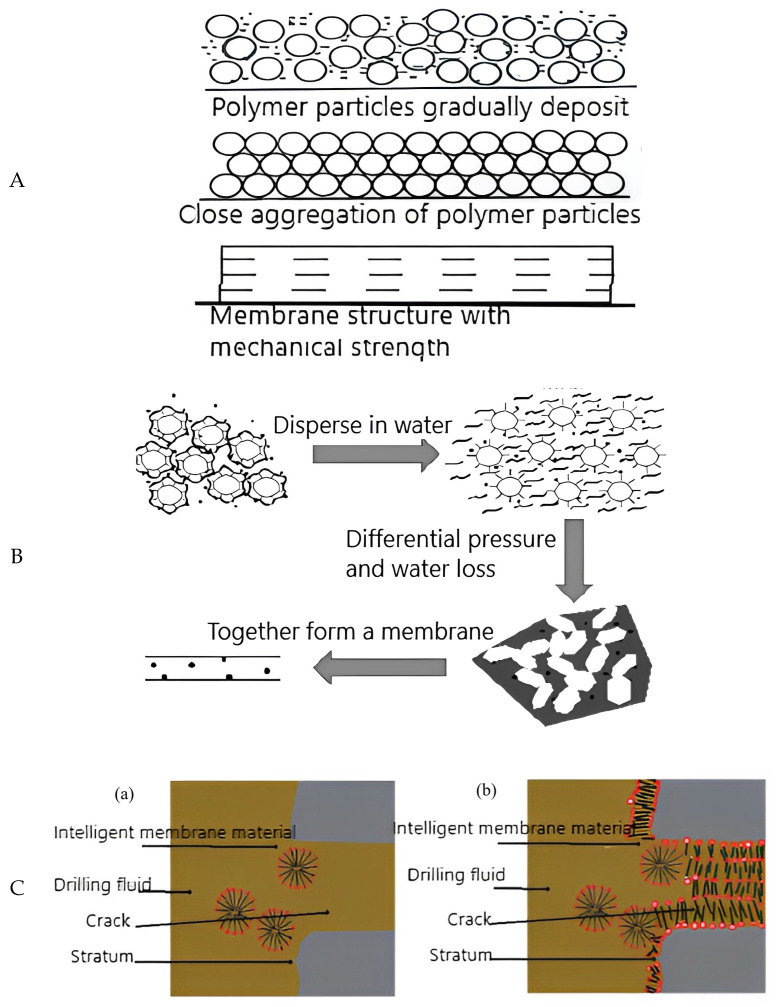
(**A**) Schematic diagram of polymer stacking into film [101]. (**B**) PF-HCM film formation process [102]. (**C**) Schematic diagram of intelligent membrane plugging principle. (**a**): Intelligent membrane material enters the leakage layer; (**b**): intelligent membrane material seals the leakage layer [38].

**Table 1 molecules-28-02975-t001:** Advantages and disadvantages of three common IPG systems and typical products.

Types of Gels	Description	Advantages	Disadvantages	Commonly Used Products
Underground gel	Gum solution (polymer/monomer + crosslinker + additives) is injected into the reservoir with the same viscosity as the polymer/monomer solution: the gum solution is gummed under reservoir conditions [48].	High injection capacity: strong flow ability in the formation, able to penetrate deep into the reservoir [49].	Gum formation is controlled by many factors, such as shear force, chromatophore adsorption, reservoir environment (pH, temperature, mineralization) and dilution, dispersion and diffusion; gum formation time is not easy to control; influenced by reservoir environment, the strength of gum formation is not easy to control; easy to enter the low permeability layer and damage the formation, may cause blockage of the low permeability formation [50,51].	Strong proprioceptive glue, weak proprioceptive glue, CDG.
Partially pre-crosslinked gel	Partial turnover prior to injection: partial pre-crosslinked gel injected into the reservoir and aged under reservoir conditions [52].	It has strong water absorption, water expansion, temperature and salt resistance, and stability. In a certain pressure, it is easy to enter the crack to produce blockage and retention; plugging agent in the crack can continue to expand, enhance the plugging effect, and improve the strength of the leakage layer [53,54,55,56,57].	Can only control part of the gel formation: part of the pre-crosslink overall more viscous, not easy to inject; gel formation process is affected by surface equipment, wellhead, and reservoir environment [58].	Body Gel.
Pre-crosslinked gels	Fully gelled before injection: fully gelled and injected into the reservoir [59].	Gel can be avoided into low permeability formations; gel is not affected by equipment and reservoir conditions at the wellhead; gel formation strength can be easily controlled [60].	Large particles of PPG cannot enter conventional porous media, and small particles cannot form an effective seal in cracks and large pore channels [61,62].	Microgel, PPG, clear water, pre-crosslinked monolithic adhesive.

**Table 2 molecules-28-02975-t002:** Temperature resistance of plugging agent.

Temperature/°C	Time Required for Gel/h	Gel Strength	Test Phenomena
70	72	H	Unchanged in 90 d
80	56	H	Unchanged in 90 d
90	48	H	Unchanged in 90 d
100	40	H	Unchanged in 90 d
110	32	H	Unchanged in 90 d
120	24	G	Mobility enhanced in 90 d

Condition: salinity was 20 × 10^4^ mg/L. H: Slightly-deformable, non-flowing gel; G: Medium-deformable, non-flowing gel.

## Data Availability

Not applicable.

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
