# Peer review of "Research Progress of Intelligent Polymer Plugging Materials"

_molecules, 2023, doi:10.3390/molecules28072975_

Round 1

Reviewer 1 Report

Interesting results and novelty work. A paper focuses on review analysis on intelligent polymer plugging materials. Though the intention of the authors is highly commendable, there is lot of problems particularly in the presentation throughout the manuscript. Besides there are many grammatical mistakes throughout the manuscript, particularly in respect of use of singular and plural with the subject or verb. In view of the above comments, whole manuscript should be properly written to make it acceptable by Molecules journal. I highly recommended this article to be accepted and published in the revised version.

 Abstract:
The abstract given here starts without any background for the present work. Of course, it contains brief details about experimental aspects and the obtained results. However this abstract does not follow the norm of an abstract, which should state briefly:

1.      The purpose of the study undertaken, what are you trying to solve
2.      brief mention of experimental aspects (without using abbreviations)
3.      highlights of the results numerically
4.      Important conclusions based on the obtained results
5.      Potential applications

Therefore, it is suggested that the Abstract to be modified as per the suggestions given above.

 Introduction

Introduction section is long with a many references based on the literature survey conducted by the authors. This is very good. However, this lacks in proper presentation of literature survey, which should have been systematic whereby existing scientific gaps should have been brought out. This should have given justification for the present study, which should be followed by the objectives of this study. In fact there is large amount of literature available on the intelligent polymer plugging materials. None of these have been brought out in this study whereby the authors have not justified why they have chosen the method they have used in their study.  It should be noted that normally 'Introduction' should give brief background through literature survey for the study citing previous published work where-by scientific gaps that exist should be brought out. This would have led to justification for the present study.  It is therefore suggested that 'Introduction Section' should be revised as suggested above because this Section is an important one from the point of view of taking up the present study.

Relevant article on polymeric materials should be cited such as https://doi.org/10.3390/ma15113852, https://doi.org/10.1080/15440478.2022.2159606, https://doi.org/10.1016/j.triboint.2023.108272.In my opinion the paper will have good merit if such applications can be demonstrated and reported. Can you give some example?

The first one is Materials which are in the field should give details of all materials used in the study, where from they were procured, known characteristics, if available (for e.g. kenaf, UP, where do you get it, what is the purity of the chemical and etc.).

 Methodology of pluging polymer, where methodologies used in the study should be given in a systematic way using sub section with numbers for each of the properties. First the processing or preparation aspects of the final material should be given followed by the characterization of prepared materials including preparation of samples for any specific property or morphology studies should be presented in a systematic way. Here one should also clearly mention the number of samples used, any standards followed for variety of properties, make and model of the instruments used for characterization, their accuracy and experimental conditions used, etc.

It should be known to the authors when one publishes any scientific paper, the results presented therein should be such they should be reproducible by any other person when the experiment is repeated using the same materials. In the present paper, it would be difficult for any other person to repeat the experiments because the chosen materials do not have any pre-history, which is required for other researchers to carryout experiments to check the possible reproducibility of the procedure adopted by these authors.

Some of the paragraph should be under results and discussion and if it is already there then this becomes repetition and hence can be deleted. Secondly, this Section is methods and hence only results should be mentioned and then it should be discussed preferably comparing it with earlier reported similar results by other researchers.

Overall discucussion is well written and easy for the reader to understand what the authors have conveyed. However, some of the paragraph should be under Methods and if it is already there then this becomes repetition and hence can be deleted. Secondly, only results should be mentioned and then it should be discussed preferably comparing it with earlier reported similar results by other researchers.

Throughout the manunscript, there are very good comparison and explanation.

Conclusions
Conclusions given here are do not reflect what had been achieved including many speculations. It is too long and should be in 1 paragraph. Hence these need to be suitably modified. It may be remembered that this Section forms a summary of all the major observations/ results obtained. Accordingly, here presentation should consist of the main Results or the observations of the study in short sentences probably with bullet points. This should stand alone or form a subsection of a Discussion or Results Section. Hence better to rewrite this Section based on the comments given in the whole text.

General Comments:
The paper though contains some interesting results and novelty work, it lacks in its proper presentation in the whole manuscript. Of course there is need for better language throughout the manuscript. It is suggested that the authors should take the help of native English speaking person to take care of this problem. In view of these, the paper is highly recommended and should be accepted for publication in the revised form. It is suggested that the authors should revise the paper in the light of above comments/suggestions.

Reviewer 2 Report

Dear Authors,

before I go through a second reading of your manuscript, please fix all the missing references: it is a prove of you have read your manuscript before submission.

After a quick lecture, the manuscript needs to be heavily rewritten for a better understanding: for example, the Abstract is really poor and you rewrite the same meaning several times.

Kind regards.

I wait for your revisions

Reviewer 3 Report

In this work, the Authors reviewed several articles reporting the development of intelligent polymer materials, especially applications in drilling fluid plugging. Generally, this manuscript requires major revisions before being considered for publication in molecules or any journal in this field

1.    Abstract, the author states that the review article also discusses the mechanism of intelligent polymers in drilling and plugging, but the text does not clearly describe this statement

2.    No explanation about SMP-LCM?

3.    The author does not clearly limit the year of publication of the cited articles

4.    Some blurry images should be increased in sharpness

5.    There should be a statement explaining that some images are reproduced or with permission from the article cited

6.    The author does not compare performance using intelligent polymers (smart polymer shape memory polymer, intelligent polymer gel, intelligent polymer membrane, and intelligent polymer bio-mimetic material) in drilling and plugging. Please fix it!

Round 2

Reviewer 2 Report

Dear authors,

the manuscript in the present form is readable, nice to read, and I would like to thank you the anonymous reviewer #1 to stress out the needed clarifications. I could not read the original version for the mess, lack of proper references, serious errors in editing, and the hole in the precision: it was rather boring to read the same sentences in few different flavors (albeit the abstract is a collection of few repetitions).

Now, I would accept as is. Congratulations.

Reviewer 3 Report

The revised article already shows a clearer summary, but there are several sentences that are ambiguous and need to be corrected, for example: [line 73-737): With low high temperature and ..............................................
